# Thick Mucus in ALS: A Mixed-Method Study on Associated Factors and Its Impact on Quality of Life of Patients and Caregivers

**DOI:** 10.3390/brainsci12020252

**Published:** 2022-02-11

**Authors:** Sarah K. Bublitz, Eva Mie, Maria Wasner, Alexander Hapfelmeier, Jens Geiseler, Stefan Lorenzl, Andrea Sylvia Winkler

**Affiliations:** 1Palliative Care Research Hub at the Institute of Nursing Science and Practice, Paracelsus Medical University, 5020 Salzburg, Austria; stefan.lorenzl@khagatharied.de; 2Department of Neurology, Agatharied Hospital, 83734 Hausham, Germany; 3General Practice, Georgenstr. 39, 80331 Munich, Germany; mieeva@web.de; 4Catholic University of Applied Sciences, 81667 Munich, Germany; maria.wasner@ksh-m.de; 5Institute of General Practice and Health Services Research, School of Medicine, Technical University of Munich, 81667 Munich, Germany; alexander.hapfelmeier@mri.tum.de; 6Institute for AI and Informatics in Medicine, School of Medicine, Technical University of Munich, 81675 Munich, Germany; 7Department of Pneumology, Sleep and Ventilation Therapy, Klinikum Vest, Paracelsus-Klinik Marl, 45657 Recklinghausen, Germany; jens.geiseler@klinikum-vest.de; 8Department of Neurology, Center for Global Health, Klinikum Rechts der Isar, Technical University of Munich, 81675 Munich, Germany; andrea.winkler@tum.de; 9Centre for Global Health, Institute of Health and Society, University of Oslo, 0373 Oslo, Norway

**Keywords:** amyotrophic lateral sclerosis, thick mucus, oral secretion, non-motor signs, quality of life, caregiver burden

## Abstract

In this explorative mixed-method pilot study, we set out to have a closer look at the largely under-recognized and under-investigated symptom of thick mucus in patients with ALS and its impact on patients and relatives. Thick mucus is a highly distressing symptom for both patients and caregivers. It complicates the use of non-invasive ventilation and is therefore an important prognostic factor of survival. Methods: In our preliminary study, we used a cross-sectional design, including ten ALS patients with thick mucus who were matched to ten ALS patients without thick mucus. Lung function tests and laboratory and sputum analysis were performed and questionnaires administered in order to determine associated factors of thick mucus accumulation. In a qualitative study using semi-structured interviews, we analysed the impact of thick mucus on patients and caregivers. Results: Reduced respiratory parameters as well as a higher degree of bulbar impairment were associated with the presence of thick mucus. Quality of life of patients and caregivers was strongly impaired by thick mucus accumulation. Conclusions: Thick mucus in patients with ALS has a strong impact on quality of life. Reduced cough flow and severely impaired bulbar function appear to be indicative parameters. We suggest that healthcare providers actively explore the presence of thick mucus in their patients and that it becomes included in commonly used screening tools.

## 1. Introduction

Amyotrophic lateral sclerosis (ALS) is a progressive and devastating neurodegenerative disease with a median survival of three years from the time of diagnosis [1]. Besides progressive muscular paralysis, symptoms include non-motor features such as pain [2], autonomic dysfunction [3], depression and excessive oral secretions including thin saliva and thick, viscous mucus. Studies in other neurodegenerative disorders, such as Parkinson’s disease, show that non-motor signs have a strong impact on the quality of life of affected individuals [4] and relatives. For several symptoms, clinical guidelines have already been established [1,5]. Since there is currently no curative therapeutic option for ALS, multidisciplinary palliative care is a central approach to alleviate symptoms, which not only affect patients but also impact caregiver burden.

Oral secretion management is a challenge for patients and caregivers. It is not only highly distressing and socially stigmatizing, but the ability to clear secretions from the upper airways is also a major factor of tolerance and effectiveness of non-invasive ventilation (NIV) [6,7]. It has been shown that a high score in the Oral Secretion Scale (OSS), reflecting the absence of oral secretions, facilitates NIV initiation and that NIV tolerance itself is an important prognostic factor of survival [8].

The aims of this pilot study were: first, to assess the clinical characteristics and risk factors of thick mucus in patients with ALS and, second, to identify to what extent the quality of life of patients and their caregivers is limited by the mucus. “Thick mucus” in this study is defined as the existence of subjective discomfort resulting from accumulating mucus in the upper airways difficult to expectorate and thus pronounced that treatment is required. Therapeutic interventions for thick mucus are not well studied and are mostly based on personal experience of neurologists or palliative care experts [9]. Excessive thin saliva is often treated with anticholinergics or botulinum toxin injections, and thick mucus sometimes even occurs as a side effect of treatment. Other patients report thick mucus in the absence of thin saliva. To some degree, thick mucus can be ameliorated by adequate hydration, the use of saline nebulisers or medications such as N-acetylcysteine or carbocysteine. As progressive cough flow weakens, respiratory physical therapy or a mechanical insufflator–exsufflator can be applied.

## 2. Materials and Methods

### 2.1. Study Design and Subjects

Twenty-two patients with the diagnosis of definite or probable ALS according to El Escorial Criteria [10,11] with or without symptoms of thick mucus and their relatives were prospectively recruited from two motor neuron disease clinics in Munich, Germany (LMU–Klinikum, Ludwig–Maximilians University and Klinikum rechts der Isar, Technical University of Munich). Atypical phenotypes and patients with cognitive decline were excluded a priori. Clinical tests and qualitative interviews were conducted at a specialized centre for respiratory medicine (Asklepios–Fachkliniken München–Gauting, Germany). Exclusion criteria were an acute upper airway infection or chronic obstructive pulmonary disease in the preceding two weeks. In addition, patients with sialorrhea based on the clinical sign of drooling in the absence of the above definition of “thick mucus” were also excluded. All patients had to be cared for by a relative who had been able to discuss the impact of thick mucus.

In a cross-sectional design, each ALS patient with thick mucus was “loosely” matched with an ALS patient without thick mucus. We hypothesized that age, sex and phenotype at onset (bulbar or limb manifestation) would be confounding variables and therefore we matched patients and controls accordingly.

In the following, we refer to ALS patients with thick mucus as “patients” and those without thick mucus as “controls”.

### 2.2. Data Collection

Demographic data and clinical examination were documented for all participating patients and controls (see Table 1) as well as for caregivers. Routine laboratory blood tests were conducted (see Table 2), as well as lung function testing including spirometry, body plethysmography, Pi.max, P0.1, peak cough flow measurement, MIC and blood gas analysis (see Table 3). Chest X-rays were performed only in the patient group. Sputum samples were collected in patients by spontaneous coughing or via use of the Cough Assist™ device and prepared for microbiological and cytological analysis performed in the Institute for Microbiology at the Asklepios Fachkliniken Munich–Gauting.

### 2.3. Questionnaires

All questionnaires were completed in person without the presence of others in the Asklepios Fachkliniken München–Gauting. One control subject and her relative was interviewed by phone because of the far distance of her home.

The questionnaires used consisted of the following (see Table 4): WHOQOL-BREF [12], HADS [13], ALS-FRS-R [14], MND Coping Scale [15], Social Withdrawal Scale [16], MDRS [17]. Additionally, we asked the following questions using Visual Analogue Scales (VAS); the first three questions were only presented to the patient group: How intense is mucus secretion? How much are you bothered by mucus secretion? How much is your quality of life influenced by mucus secretion? How would you rate your level of anxiety?

### 2.4. Qualitative Interviews with Patients and Caregivers

Semi-structured interviews (*n* = 10) were conducted with all patients with thick mucus and with one caregiver for each of them. Verbal communication was not possible for four subjects in the patient group, but three of them were capable of written communication. One patient was unable to speak or write; thus, her husband assisted in nonverbal communication.

The opening question to patients was “How does mucus secretion affect your daily life and your quality of life?” Relatives were asked: “To what extent does mucus secretion affect your own quality of life and the daily life of both of you? Please try to estimate how mucus congestion affects the quality of life of your relative.” An interview guideline was used in the further course of the interview. The interviews were tape recorded and transcribed by the interviewer verbatim. Transcribed interviews were analysed via qualitative content analysis [18]. Transcripts were screened for significant statements related to the central topic of thick mucus secretion influencing quality of life, categorized and grouped together to identify common themes.

### 2.5. Statistical Analysis

The program IBM SPSS Statistics (IBM Corp., Armonk, NY, USA) was used for statistical analysis. The distribution of continuous variables is described by mean ± standard deviation or median and range, depending on the assumption of a normal distribution. Accordingly, *t* tests and Mann–Whitney U tests were applied for hypothesis testing of group differences. A multivariable linear regression model was used to account for potential confounding. Hypothesis testing was performed at exploratory two-sided 5% significance levels.

## 3. Results

### 3.1. Qualitative Data

#### 3.1.1. Patient Characteristics

A total of 11 ALS patients with thick mucus and their caregivers were recruited. One patient had to be excluded from the trial due to acute pneumonia. Median age and time since diagnosis were comparable in both groups. Four patients, but none of the controls, used percutaneous endoscopic gastrostomy (PEG) for nutrition (Table 1).

**Table 1 brainsci-12-00252-t001:** Characteristics of patients and controls.

		Patients (*n* = 10)	Controls (*n* = 10)	*p* Value
Female		6	6	---
Male		4	4	---
Age	Median (range)	70 (56–76)	67.5 (48–82)	0.819
Site of onset: limb		7	7	
Site of onset: bulbar		3	3	
Time since diagnosis (months)	Median (range)	25 (11–192)	25.5 (14–151)	0.739
ALS-FRS-R total	Median (range)	25.5 (9–37)	38.5 (24–44)	0.001
ALS-FRS-R sub-scores				
Bulbar function	Median (range)	7 (0–10)	11.5 (3–12)	
Fine motor function	Median (range)	5 (0–11)	9 (6–12)	
Gross motor function	Median (range)	4 (0–10)	7 (4–12)	
Respiratory function	Median (range)	8.5 (6–11)	12 (8–12)	
PEG	Total number	4	0	---
NIV	Total number	8	3	---

ALS-FRS-R: ALS functional rating scale revised, PEG: percutaneous endoscopic gastrostomy, NIV: non-invasive ventilation.

The ALS-FRS-R total score and sub-scores differed significantly between both groups (Table 1). Comparison of the ALS-FRS-R sub-scores showed that patients overall performed worse than controls. Only two patients showed normal ability to speak, compared to six in the control group. One patient, but seven controls had no sialorrhoea at all. Concerning dyspnoea, all control subjects scored three points or higher (dyspnoea occurs when walking or none), whereas dyspnoea occurred in two patients already at rest or when eating or dressing. Four patients used a scopolamine transdermal patch to treat pseudohypersalivation, one patient took amitriptyline and one patient trimipramine at the time of examination. Past or current treatment efficacy with regard to thick mucus was not documented.

#### 3.1.2. Clinical Examination

There was no relevant difference in heart rate, blood pressure, body temperature and BMI. Respiratory rate was mildly elevated in patients.

#### 3.1.3. Laboratory Results

Results from laboratory testing showed no relevant differences among the study groups (Table 2). In particular, no signs of infection or dehydration were detected.

**Table 2 brainsci-12-00252-t002:** Laboratory results (excerpt).

		Patients	Controls	*p* Value
Haemoglobin	Median (range)	14.3 (11.6–15)	13.9 (11.4–16.5)	0.85
Haematocrit	Mean (standard deviation)	0.41 (+/−4.18)	0.41 (+/−2.57)	0.98
Leucocytes (G/L)	Mean (standard deviation)	7.2 (+/−1.6)	7.3 (+/−2.2)	0.9
Creatinine (mg/dL)	Median (range)	0.7 (0.2–2.1)	0.8 (0.7–1.1)	0.06
CRP (mg/dL)	Median (range)	1.3 (0.1–67.8)	1.9 (0.5–57.9)	0.18

#### 3.1.4. Spirometry and Cough Flow

Vital capacity (VC) and forced expiratory volume in 1 s (FEV1) were significantly reduced, and forced vital capacity (FVC) and peak expiratory flow (PEF) were borderline significantly reduced in the patient group (Table 3). Bicarbonate and BE suggested that patients had to compensate for respiratory acidosis more than controls, and peak cough flow (PCF) and maximum insufflation capacity (MIC) were borderline reduced in the patient group. Other results from spirometry and body plethysmography, as well as the results of blood gas analysis, showed no relevant differences.

**Table 3 brainsci-12-00252-t003:** Results from spirometry and cough flow.

		Patients	Controls	*p* Value
VC (% pred.)	Mean (standard deviation)	61.5 (+/−24.9)	95.1 (+/−22.5)	0.008
FVC (% pred.)	Median (range)	66 (31.4–116.8)	85.6 (55–135.5)	0.07
FEV1 (% pred.)	Mean (standard deviation)	62.5 (+/−27.2)	99.2 (+/−21.5)	0.006
FEV1/FVC (% pred.)	Mean (standard deviation)	96.1 (+/−10.8)	97.1 (+/−15.8)	0.89
Resistance (% pred.)	Mean (standard deviation)	79.5 (+/−34.1)	91.7 (+/−46.4)	0.53
P0.1 (% pred.)	Median (range)	138.4 (53–284.5)	138.1 (63–191.9)	0.55
Pi.max (% pred.)	Mean (standard deviation)	29.2 (+/−12.1)	38.0 (+/−16.5)	0.24
PEF (% pred.)	Mean (standard deviation)	52.0 (+/−22.9)	86.9 (+/−21.4)	0.08
pO_2_ (mmHg)	Mean (standard deviation)	72.4 (+/−5.5)	72.8 (+/−8.8)	0.41
pCO_2_ (mmHg)	Median (range)	37.9 (33.2–56.1)	36.4 (32.9–40.1)	0.16
SO_2_ (%)	Median (range)	94.6 (92.4–96.9)	95.6 (92.3–97.1)	0.45
Bicarbonate (mmol/L)	Mean (standard deviation)	27.9 (+/−4.1)	25.0 (+/−1.0)	0.08
BE (mmol/L)	Median (range)	3.5 (−0.3–6.8)	1.9 (−1.1–4.0)	0.08
PCF (L/min)	Median (range)	150 (70–200)	260 (200–420)	0.08
MIC (mL)	Median (range)	2700 (1200–3500)	3200 (2500–5710)	0.08

VC: vital capacity, FVC: forced vital capacity, FEV1: forced expiratory volume in one second, FEV1/FVC: relative one-second capacity, P0.1: mouth occlusion pressure at 100 ms during quiet breathing, Pi.max: maximal inspiratory mouth pressure, PEF: peak expiratory flow, pO_2_: partial pressure of oxygen, pCO_2_: partial pressure of carbon dioxide, SO_2_: saturation of oxygen, BE: base excess, PCF: peak cough flow, MIC: maximum insufflation capacity.

#### 3.1.5. Sputum Analysis

Sputum was retrieved from 7/10 patients. In three patients, the collection of sputum samples could not be accomplished, neither by coughing nor by using the Cough Assist™ device. Mucus was not detected in 3/7 samples; in the other samples, there was a small or modest amount of mucus. All samples contained squamous epithelium, alveolar macrophages and lymphocytes. Basal cells, generally found in the presence of erosions and ulcerations of bronchial mucosa, were not detected. The detection of alveolar macrophages proved that the analysed sputum originated from lower airways and was not only saliva. Ciliated epithelium was only found sporadically in 1/7 samples. The number of ciliated cells can be increased in acute bronchitis [19]. Sporadic or moderate amounts of neutrophilic granulocytes were found in 6/7 sputum samples; in one sample, no neutrophilic granulocytes were detected. Six samples underwent microbiological analysis. A high number of bacteria from the oral flora was detected in all six samples. Candida albicans was present in small amounts in 3/6 samples and in high amounts in 1/6 sample. Small numbers of Haemophilus influenzae were detected in one sample. This patient did not show any signs of systemic infection. Food residues were not detected in the samples.

#### 3.1.6. Questionnaires

Results from the questionnaires WHOQOL-BREF, HADS, MND Coping Scale, and Social Withdrawal Scale did not differ significantly between the two groups, although coping capacity and quality of life were reduced, and social withdrawal occurred more often in the patient group (Table 4).

**Table 4 brainsci-12-00252-t004:** Results from questionnaires.

		Patients	Controls	*p* Value
WHOQOL-BREF (%)	Mean (standard deviation)	65.5 (+/−9.46)	71.2 (+/−9.65)	0.22
HADS (%)	Mean (standard deviation)	36.5 (+/−14.27)	32.2 (+/−14.37)	0.53
Coping (%)	Median (range)	69 (52–75)	74.5 (57–77)	0.18
Withdrawal (%)	Mean (standard deviation)	24.44 (+/−3.24)	21.8 (+/−5.19)	0.22
MDRS (%)	Median (range)	45.5 (20–61)	* ---	---

* Only one control subject indicated dyspnoea and completed the MDRS (score 55). A comparison of this scale between the two groups was therefore not performed. WHOQOL-BREF: World Health Organization Quality of Life assessment, abbreviated version; 26 questions with scores between 26 and 130; higher scores reflecting higher quality of life. HADS: Hospital Anxiety and Depression Scale, 7 questions regarding anxiety and 7 questions regarding depression with scores between 0 and 3 for each question; higher scores reflecting a high degree of anxiety/depression. Coping: MND Coping Scale; 22 items with scores between 1 and 6 for each item; higher scores reflecting higher coping capacity. Withdrawal: Social Withdrawal Scale; 24 items with scores between 0 and 3; lower scores reflecting higher degree of social withdrawal. MDRS: MND Dyspnea Rating Scale, scores between 0 and 44, higher scores reflecting high degree of dyspnoea.

A regression analysis was carried out in order to adjust the group comparison of WHOQOL-BREF by ALS-FRS-R. The mean baseline WHOQOL-BREF was reduced by 5.72 points for patients compared to controls. After adjustment, the mean WHOQOL-BREF score was increased by 2.61 points in the patient group (patients: 69.66%, controls: 67.04%).

Results from the VAS questions showed that the level anxiety was comparable in both groups (Table 5). Subjective intensity of thick mucus and the degree of how much the patient was affected by it were fairly variable but overall showed that thick mucus is an important factor influencing quality of life in ALS.

**Table 5 brainsci-12-00252-t005:** Results from Visual Analogue Scale questions.

		Patients	Controls	*p* Value
How would you rate your level of anxiety?	Median (range)	32.5 (0–75)	27.5 (15–50)	0.939
		Patients only		
How intense is mucus secretion?	Median (range)	40 (20–100)		
How much are you bothered by mucus secretion?	Median (range)	50 (20–100)		
How much is your quality of life influenced by mucus secretion?	Median (range)	60 (30–80)		

Visual Analogue Scale (VAS; range 0–100).

### 3.2. Qualitative Data, Problem-Oriented Interviews of Patients and Caregivers

Analysis of the transcribed interviews identified the following five central themes.

#### 3.2.1. Attempt to Find an Explanation

Many patients and caregivers seek an explanation for thick mucus secretion. Many remember a preceding cold; others see it in context with pseudohypersalivation, reflux, or dehydration.

#### 3.2.2. Physical Effects

Patients describe a consistent foreign body sensation using different metaphors (“like a dumpling in the throat which does not dissolve”). They feel a constant need to clear their throat, but without success due to weakened cough flow. Some patients describe sudden, severe coughing attacks (“the feeling that I am ripped into pieces”) and that they afterwards feel exhausted for hours. Thick mucus can result in choking and in a feeling of suffocation that can be frightening for both patients and caregivers, especially when this occurs during the night. For many patients and caregivers, thick mucus leads to lack of sleep. (patient: “I often sleep for 15 minutes, then I wake up again and see if there is thick mucus, then at some point I fall asleep again. Nights are terribly long. Very long”).

#### 3.2.3. Everyday Life

Due to mucus accumulation, patients often can no longer be on their own. This is a burden for everybody involved and leads to a sleep deficit for patients as well as caregivers (patient: “somebody has to watch me also during the night, because when I have thick mucus, then I have to be suctioned. So, somebody has to be there all the time”). Only few patients report that mucus secretion is embarrassing for them and a reason for social withdrawal.

#### 3.2.4. Psychological Effects on Patients

The most serious psychological effect is fear. Patients are afraid to choke or develop pneumonia due to aspiration (patient: “You are afraid that something gets deeply into the lung and that one develops pneumonia. There could be some food in the mucus. This would be a death sentence. You would not be able to remove it from the lung. (…) because if I choke very badly, then I would need urgent suctioning and it would take too long for them to get here from the hospital.”). Other patients feel constrained by mucus secretion, are stressed out by it and feel nervous. Thick mucus reflects progression of the disease, which is considered burdensome (patient: “I always say, you get used to a lot. Sitting in a wheelchair. You can get used to it. Of course, it would be nicer, if one could move the upper body parts. And I could content myself with this (thick mucus). But because it constantly progresses, that shows it unmistakably, that’s it”).

#### 3.2.5. Psychological Effects on Caregivers

The psychological burden for caregivers associated with thick mucus originates mainly from helplessness and fear (caregiver: “I think thick mucus is the worst aspect, because this is the only situation in which I am unable to help. Yes, I can help lifting him, I can…but if I see this, I just don’t know what to do, because I only see the eyes and he cannot breathe, looks at me and I cannot do anything…This is a lack of power, the worst thing, you can just be there. Yes. In our whole life, we are used to always do something and then you get into situations where you cannot do anything. This helplessness…”). In addition, to caregivers, thick mucus reflects progression of the disease. They have to deal with how to react, for example in terms of home care (caregiver: “She is stepping over into the next stadium, this is my feeling at the moment, this is it with the mucus, because this is all new and really since yesterday it has become even worse. So, there is fear, not acute, like she will choke to death, more like: how will we proceed from here, what are the consequences and how can we help her”).

## 4. Discussion

Thick mucus accumulation in ALS patients is under-investigated and still poorly understood. Moreover, the impact on caregiver burden has not been investigated previously. Therefore, we initiated a small case-control pilot study to identify potential causes and the clinical impact of thick mucus. Laboratory testing and chest X-ray showed no signs of infection, dehydration, pneumonia or congestive heart failure.

A main focus of our qualitative pilot study was on the problem-centred interviews which we performed to analyse psychosocial aspects of thick mucus accumulation in ALS. This interview form is often considered challenging in patients not able to communicate verbally. Some of our patients were able to write but, in most cases, used only few words to outline their situation. Still, we were able to identify several relevant statements from patients and caregivers, whose quality of life often seems affected considerably by the patients’ symptoms. Central aspects of our interviews were the fear of choking and disease progression for patients and caregivers, as well as helplessness. Whereas the fear of choking was often mentioned in the interviews, results of the anxiety part of the HADS did not correlate with this. Patients described this fear as selective and depending on the situation, which might not be covered well by common questionnaires. Importantly, quality of life in ALS patients has been shown to be significantly influenced by fear [20]. Fear was also a central topic for caregivers, as it leads to helplessness and affects their quality of life. Many caregivers appeared to be overburdened with the symptom of thick mucus and did not dare to leave the patient alone. Mainly, the fear that thick mucus triggers choking to death has been mentioned by caregivers, which increased the burden significantly. Furthermore, the appearance of thick mucus might trigger the next step of the disease in their opinion. Therefore, we would like to emphasize that thorough explanation of the symptom origin and addressing and alleviating the fear of choking should include patients as well as caregivers. Unfortunately, treatment options are mainly based on the recommendations for sialorrhea in ALS and focus on reducing the production of saliva [21], which are often not sufficient. In some patients, mucus can only be removed mechanically by use of a suction device.

Furthermore, we investigated the sputum in the patient group. Sputum, in general, is easily accessible and represents a mixture of upper and lower secretion. Mucus was detected only in mild to modest amounts, or not at all, in the patient group and there were no signs of acute infection. It was, however, difficult to gather enough sputum from all patients, as bronchoscopy with bronchoalveolar lavage (BAL) was considered too invasive. This procedure should, if possible, be included in future studies in both patients complaining of thick mucus and in controls.

We assume that thick mucus in ALS neither results from excessive mucus production, nor is it a result of airway infection. Rather, it appears to be a consequence of decreased elimination by impaired clearance of the airways. The two physiological ways of airway clearance are mucociliary clearance and coughing. Cilia movement in the mucosa enables transport of mucus, liquid and inhaled particles to proximal airways. Coughing involves the coordinated activation of several muscles of the upper airways, chest and abdominal wall and is often impaired in ALS. Upper airway muscles of the bulbar region greatly influence airflow. Their activation needs to be synchronized with the respiratory muscles to produce cough flow. The force developed during the expulsions of cough (“peak cough flow”) largely depends on the magnitude of the preceding inspiration [19]. Results from lung function testing in our case-control study showed reduced selected respiratory parameters (Table 3) and overall worse ALS-FRS-R sub-scores in the patient group (Table 1). We conclude from our data that mucus accumulation in ALS patients results mainly from impaired airway clearance by weakened cough. When considering this mechanism, thick mucus could also be labelled as a motor sign, rather than a non-motor sign in ALS, whereby the approach of whether this is classified as a motor or non-motor sign is inconsistent in recent reviews [22,23].

Although the widely used functional scale ALS-FRS-R does not include data about thick mucus [14], patients still had a significantly lower score, reflecting a higher degree of physical impairment. However, the result has to be interpreted carefully as patients were matched for disease phenotype at symptom onset. It can be assumed that without matching, the difference between the two groups would have been even more pronounced. Quantitative measures of quality of life used in this study included established questionnaires and VAS. The WHOQOL-BREF showed that quality of life was mainly associated with physical limitations as opposed to psychological or social factors. In the VAS questions, patients stated clearly that their quality of life had declined due to thick mucus. This finding is similar to the effect of other non-motor symptoms in ALS [2,3]. We believe that non-motor symptoms in ALS are not covered well by standard screening tools and need to be investigated further.

A clear drawback of our pilot study is the small number of patients. We tried to account for the potential risk of confounding by matching for already known confounding variables in this cross-sectional study, i.e., age, sex and phenotype at onset (bulbar or limb manifestation). In retrospect, matching should have included other confounders suggesting an advanced stage of the disease, such as pulmonary function testing, and NIV/PEG dependence. However, the ideal approach for an analysis of risk factors would be a longitudinal cohort study and an assessment of all potentially contributing factors. In addition, the definition of thick mucus could be improved and preferably based on clinical signs/symptoms, as well as histology of sputum. Differentiation from sialorrhea must be clear, and scales for the assessment of both symptoms and their severity should be designed and implemented to measure treatment effects, as suggested by Garuti et al. [21].

In summary, this study has its main merit as a qualitative pilot study exploring patient and caregiver perspectives of thick mucus in ALS. Our results suggest that reduced respiratory parameters and a higher degree of impairment in bulbar function contribute to thick mucus in ALS patients. Subjective quality of life is impaired, and therapeutic options are scarce, but response to symptomatic treatment, which was not evaluated in our study, is likely to have a significant impact on quality of life. However, there were no significant differences between the patient and control group in the more objective assessment of quality of life and psychiatric comorbidity using questionnaires, which could be due to the unbalanced study design and the small sample size. Therefore, larger, possibly multicentre studies with larger group sizes should be designed and conducted that take into account the pitfalls highlighted in our small pilot study. Similar to other non-motor signs in ALS, healthcare providers should actively ask for the symptom of thick mucus and explain pathogenesis as well as treatment options. Common screening tools and questionnaires need to be updated accordingly.

## Data Availability

Data can be made available upon request.

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
