# Peer review of "Thick Mucus in ALS: A Mixed-Method Study on Associated Factors and Its Impact on Quality of Life of Patients and Caregivers"

_brainsci, 2022, doi:10.3390/brainsci12020252_

Round 1

Reviewer 1 Report

This is an important study by Bublitz et al describing the impact of thick mucus on the quality of life of patients with ALS. It is an explorative mixed-method study designed to examine the largely under-recognized and under-investigated symptom of thick mucus in patients with ALS and its impact on patients and relatives. Thick mucus was defined as the existence of subjective discomfort based on accumulating mucus in the upper airways that was difficult to expectorate.

  • The major concern of this study is that the control group was overall healthier than the “patient” study group. The patients had worse PFTs, required more NIV and PEG dependence. As a result the impact on quality of life could be confounded by these factors. Ideally it would have been nice to match the controls and patients with the severity of PFTs.
  • The authors should mention that PFTs, bulbar pathology, etc… should be matched so that one can truly assess the impact of thick mucus without other confounders.
  • Sputum was not gathered from all patients so was not assessed. Was sputum analyzed in controls? A bronchoscopy and BAL might be beneficial to truly assess a compare the sputum in both groups.
  • Finally, a minor concern – the authors state that this is non-motor . However it is stated that the thick secretions result from the lack of ability to cough which requires motor neurons and is therefore a motor complaint since motor control of the diaphragm and upper airways is needed to clear secretions.

Reviewer 2 Report

In this study, the authors explore the symptom of thick mucus in ALS. It is already well recognised that thick secretions are a disabling non-motor symptom of ALS and its awareness and management is also standard in any specialist ALS multidisciplinary clinic (for example, Garuit et al 2019; EFCN clinical guidelines). Similarly, the cause of this symptom, i.e., largely due to pharyngeal and respiratory muscle weakness, is also well referenced in the literature and understood in practice. The interest in this study therefore lies in understanding the impact of this symptom on QoL, and the reviewer's comments focus on this aspect. As their use of qualitative interviews provides only subjective information, the authors are encouraged to further develop this aspect of the study to provide useful insights for clinical practice.

Main suggestions:

  1. The study aims/objectives and hypothesis are missing from the introduction. With relevance to this, why were the respiratory and blood analyses conducted- was there an alternate hypothesis regarding the aetiology of thick mucus, or was this done for exclusion of contributory pathology?
  2. Anterior secretions (thin) need to be clearly differentiated from posterior secretions (thick mucus) for this very specific analysis. What was the definition of sialorrhea- was it the presence of thin saliva alone? The authors define thick mucus as “subjective discomfort based on accumulating mucus in the upper airways difficult to expectorate and so pronounced that treatment was required”. How did the authors specify/emphasise ‘thickness’ of this accumulation, so as to avoid overlap with thin saliva accumulation? How was the severity/level of thick mucus analysed, as presumably this would also impact QoL assessments?
  3. Unsurprisingly, the patients with thick mucus are characterised by more disability and more bulbar involvement, with the majority already on NIV (8/10 vs 3/10 in control patients) to suggest they are at a later stage of disease. The level and severity of disability would be of influence on QoL assessment, and more detailed information needs to be provided to understand comparison between these groups. What were the El Escorial Criteria category (all definite?), what was the site-of-onset breakdown for all participants, what were the ALSFRS-R subscores (bulbar, fine and gross motor, respiratory)? The latter, particularly the bulbar subscore, is more useful than one combined score across these multidimensional domains. Given the small participant numbers, Table 1 should detail these features for every participant.
  4. Given the authors’ definition of patients with thick mucus, the patient group (n=10) must all be on treatment for this symptom but only 3 are mentioned (section 3.1.1 lines 142-143)? Treatment response would clearly be of influence on perceived impact of this symptom on QoL. Was this factor assessed and incorporated into the analysis? This information (i.e., treatment type and efficacy) should also be presented in full (in Table 1 or separately).
  5. Exclusion criteria: were atypical phenotypes (e.g., PBP, PMA, PLS) and participants with cognitive involvement excluded? The latter would impact interpretation of questionnaires and the former adds to the heterogeneity of disease experience and QoL. As cognitive involvement particularly manifests in patients with bulbar-onset disease, this is relevant to define.
  6. It is unsurprising that when specifically questioned, patients will attest to the presence of thick mucus as influential to their quality of life, yet there were no significant differences in the QoL questionnaires between patients and controls. The overall lack of impact from this symptom would be an important and more relevant conclusion.
  7. What clinically relevant points or change to practice can the authors provide from the central themes identified in section 3.2? For example, other than better treatments, what do patients/carers want better addressed regarding this symptom- better information provision on symptoms, on treatment or consequences? More discussion to alleviate the fear of choking? More input from allied health support?

Round 2

Reviewer 2 Report

Overall, the authors’ revisions have improved the readability of the manuscript. The major point of unresolved critique (as highlighted by both reviewers) is that the unbalanced study design (e.g., baseline differences in functional disability) does not allow for an unbiased interpretation of QoL. The authors have also not fully addressed the influence of therapeutic efficacy- although the lack of this information is confirmed in the authors’ responses, it does not seem to have been addressed in the revised manuscript. In the setting of the current limitations, improving the robustness of the objective conclusions requires a revised study design (e.g., matched cohorts, larger group sizes and a longitudinal approach). The paper has its main merit as a qualitative pilot study exploring patient and carer perspectives of this specific symptom.
